# Adipose-Derived Lipid-Binding Proteins: The Good, the Bad and the Metabolic Diseases

**DOI:** 10.3390/ijms221910460

**Published:** 2021-09-28

**Authors:** Laurie Frances, Geneviève Tavernier, Nathalie Viguerie

**Affiliations:** 1Inserm, UMR 1297, Institute of Metabolic and Cardiovascular Diseases (I2MC), 31432 Toulouse, France; laurie.frances@inserm.fr (L.F.); genevieve.tavernier@inserm.fr (G.T.); 2Paul Sabatier University, Université de Toulouse, 31330 Toulouse, France

**Keywords:** metabolic syndrome, diabetes, cardiovascular disease, adipose tissue, adipokine, calycin, lipocalin, apolipoprotein

## Abstract

Adipose tissue releases a large range of bioactive factors called adipokines, many of which are involved in inflammation, glucose homeostasis and lipid metabolism. Under pathological conditions such as obesity, most of the adipokines are upregulated and considered as deleterious, due to their pro-inflammatory, pro-atherosclerotic or pro-diabetic properties, while only a few are downregulated and would be designated as beneficial adipokines, thanks to their counteracting properties against the onset of comorbidities. This review focuses on six adipose-derived lipid-binding proteins that have emerged as key factors in the development of obesity and diabetes: Retinol binding protein 4 (RBP4), Fatty acid binding protein 4 (FABP4), Apolipoprotein D (APOD), Lipocalin-2 (LCN2), Lipocalin-14 (LCN14) and Apolipoprotein M (APOM). These proteins share structural homology and capacity to bind small hydrophobic molecules but display opposite effects on glucose and lipid metabolism. RBP4 and FABP4 are positively associated with metabolic syndrome, while APOD and LCN2 are ubiquitously expressed proteins with deleterious or beneficial effects, depending on their anatomical site of expression. LCN14 and APOM have been recently identified as adipokines associated with healthy metabolism. Recent findings on these lipid-binding proteins exhibiting detrimental or protective roles in human and murine metabolism and their involvement in metabolic diseases are also discussed.

## 1. Introduction

Obesity is still a major public health concern since almost half a century, and the global pandemic is overwhelming every country, even the least industrialized countries. In a 2015 study collecting obesity data from 195 countries, the Global Burden of Disease 2015 Obesity group estimated that 603 million adults and 107 million children were obese; meaning that, all over the world, one human being in 10 is obese [1].

Obesity is characterized by an excess of adipose tissue, originated from an imbalance between energy intake and energy expenditure [2]. Besides its prominent role in fatty acids trafficking, adipose tissue releases a wide range of factors called adipokines with local paracrine, and broad systemic endocrine effects on whole body homeostasis. In obese individuals, secretion of a majority of adipokines is increased. Among those, several promote inflammation and insulin resistance, which participate in the onset of cardiovascular diseases, diabetes and cancer. Till now, only a few adipokines display low levels in obesity, some having atheroprotective, anti-inflammatory and insulin-sensitizing properties such as adiponectin, the most well-known beneficial adipokine [3].

Among this swarm of factors, six are lipid-binding adipokines: Retinol binding protein 4 (RBP4), Fatty acid binding protein 4 (FABP4), Apolipoprotein D (APOD), Lipocalin-2 (LCN2), Lipocalin-14 (LCN14) and Apolipoprotein M (APOM). These lipid-binding proteins are all part of the calycin superfamily, a family of proteins with structural similarities, which is composed of the lipocalin family, the fatty acid binding protein family, the avidin family and the metalloproteinase inhibitor family [4,5].

The lipocalin family was named by Pervaiz and Brew in 1987 and is composed of small extracellular proteins, generally of 160–180 amino acids, with a sequence homology barely reaching 20%. All members of this family, including RBP4, LCN2, LCN14, APOD and APOM, share a conserved tertiary structure formed by a β-barrel of eight β-strands linked by seven loops (Figure 1). The family is divided in two groups, kernel and outlier lipocalins [5]. Kernel lipocalins like RBP4, LCN2 or APOD have three highly conserved structural regions, outlier lipocalins like APOM and LCN14 only have one or two of these conserved regions [5].

The fatty acid binding proteins (FABPs) family is composed of nine proteins of approximately 15 kDa. All of them are intracellular proteins of 126–137 amino acids [6]. They were firstly identified in liver (FABP1), intestine (FABP2), heart and skeletal muscle (FABP3), adipose tissue (FABP4), epidermis (FABP5), *ileum* (FABP6), brain (FABP7), peripheral nervous system (Myelin P2/FABP8) and testis (FABP9), but they show no tissue-specificity. FABPs share a conserved tertiary structure formed by a β-barrel of 10 antiparallel β-strands (Figure 1). They also present two α-helixes (named A1 and A2) forming a lid over the β-barrel [7].

For lipocalins and FABPs, this specific ‘coffee-cup’ structure forms a hydrophobic pocket and allows binding to small lipidic molecules. RBP4, FABP4, APOD, LCN2, LCN14 and APOM ligands and main biological functions are summarized in Table 1.

RBP4 and FABP4 could be considered as detrimental to metabolism because they exhibit high circulating levels and adipose tissue gene expression with obesity related disorders. APOD and LCN2 are ubiquitously expressed proteins with deleterious or beneficial effects, depending on their site of expression. LCN14 and APOM are newly identified adipokines associated with healthy adipose tissue and metabolism.

In this review, we will focus on these six adipokines and their implication in metabolic disorders such as obesity and diabetes.

## 2. RBP4

RBP4 was identified in the 1960s by Kanai et al. as the plasma transporter of retinol complexed with transthyretin (TTR). This 21 kDa lipocalin is mainly expressed in liver and adipose tissue, and various other tissues at lower levels, and is the main—if not the only—transporter of vitamin A in vivo [8]. Some gain-of-function mutations on *RBP4* promoter were reported to be associated with a higher risk of developing obesity or type 2 diabetes [15,16] while a non-coding mutation in *RBP4* promoter is associated with dyslipidemia [17]. Two specific membrane receptors for RBP4 have been discovered so far: the signaling receptor and transporter of retinol 6 (STRA6), expressed widely but not in liver, and the liver-expressed retinol-binding protein receptor-2 (L-RBPR or RBPR2) [18,19].

Several publications report (i) high RBP4 concentration in plasma or high levels of *RBP4* mRNA in adipose tissue of obese, diabetic individuals and their close relatives [20,21]; (ii) a positive association between serum RBP4, insulin resistance and adiposity [20,22]; (iii) a negative association (or inverse correlation) between serum RBP4 and HDL levels [20]. Similar results are observed in obese children [23]. Others also report a decrease in circulating RBP4 associated with an increase of insulin sensitivity during weight loss, subsequent to exercise training, low calorie diet or gastrectomy [20,22]. Moreover, RBP4 plasma concentrations are positively associated with blood pressure and atherosclerotic plaque in obese patients [24]. Additional recent reports in humans are reviewed in [8,9].

Furthermore, *RBP4* gene expression in human adipose tissue and heart positively correlates with expression of inflammatory markers such as the mannose receptor C-type 1 (MRC1), C-C motif chemokine ligand 18 (CCL18) or Toll-like receptor 4 (TLR4), and with cardiac hypertrophy markers such as the natriuretic peptides [25,26]. RBP4-induced TLR4 expression activates the NLRP3 inflammasome in mice, which leads to the expression of inflammatory cytokines such as IL-1β, promotes insulin resistance and cardiac hypertrophy [26,27]. Animal models give further insight into the role of RBP4 in obesity and diabetes. Adipose *Glut4*-deficient diabetic mice display elevated *Rbp4* mRNA level in adipose tissue and increased plasma RBP4, as well as *ob*/*ob* mice and mice fed with a high fat diet (HFD) [28]. Mice overexpressing human or murine *RBP4* in adipocytes and fed with HFD are more prone to develop obesity, insulin resistance and hepatic steatosis than wildtype littermates [28,29]. These mice also have higher levels of inflammatory markers like tumor necrosis factor α (TNFα) [29]. Mice overexpressing *Rbp4* in blood vessels musculature exhibit a higher blood pressure and an impaired eNOS-mediated vasodilatation [30]. Interestingly, these results could only be obtained in adipose tissue and muscle, as mice overexpressing *Rbp4* specifically in liver do not display an impaired glucose metabolism, suggesting that RBP4 might have different roles in hepatic, adipose and vascular functions [31].

In several studies, insulin sensitivity in mice was improved when targeting RBP4. When administrated fenretinide, a synthetic retinol derivate which prevents RBP4 binding to TTR, rosiglitazone, or TTR antisense oligonucleotides which lower TTR levels and induce RBP4 clearance, insulin sensitivity in mice was restored [28,32].

So far, RBP4 has been presented as a deleterious adipokine which promotes insulin resistance, obesity and cardiovascular diseases. Even if the RBP4 signaling is not entirely deciphered, many publications tend to demonstrate the negative effect of RBP4 on metabolism and how it can be a promising therapeutic target. In addition, a recent study from Fenzl et al. showed that RBP4 is critical in humans and mice for adipose adaptative thermogenesis during a cold exposure [33]. This discovery highlights new perspectives on RBP4 role in adipose tissue and metabolism, maybe not as detrimental as it first appeared.

## 3. FABP4

Also called the adipocyte lipid-binding protein (ALBP), adipocyte FABP (A-FABP) or adipocyte protein 2 (aP2), FABP4 is a 15 kDa protein that binds fatty acids and structurally related lipids. It is mainly expressed in mature adipocytes, macrophages and capillary endothelial cells. In adipocytes, this calycin protein is located in the cytoplasmic compartment of the adipose cell where it interacts with the adipose triglyceride lipase (ATGL) co-activator CGi-58 and the hormone-sensitive lipase (HSL) to enhance lipolysis [10]. It is secreted in the blood stream and the adipocyte is the prevailing contributor to circulating levels [10].

In humans, serum levels of FABP4 are higher in obese, type 1 diabetic, type 2 diabetic and gestational diabetic individuals compared to non-obese, non-diabetic subjects [34,35,36,37]. Gastric bypass surgery reduces circulating FABP4 by 42% in obese patients with type 2 diabetes, whereas behavioral and nutritional intervention alone does not reduce the serum levels of FABP4 [35]. Cohort studies including obese or diabetic patients with their family revealed that FABP4 levels were high with first-degree relatives of type 1 diabetic patients [37] and in children of obese parents, even if they were not obese themselves [38]. FABP4 plasma levels are positively associated with BMI, waist circumference and metabolic syndrome, but also with inflammatory markers like CRP or IL-6 in type 2 diabetic subjects [39]. Taken together, these observations lead to the conclusion of a detrimental role of FABP4 on metabolism. Interestingly, a recent study focusing on type 1 diabetic patients revealed that low FABP4 plasma levels, due to a functional, low-expression variant in the gene promoter, was associated with a 2.4-fold higher risk of cardiovascular disease [40]. Further comparative studies on the role of FABP4 between type 1 and type 2 diabetes are required to elucidate these differences.

Adverse effects of FABP4 in patients with obesity, metabolic syndrome or diabetes are supported by in vitro and in vivo observations. Mice with diet or genetically induced obesity and lacking FABP4 have lower plasma glucose, triacylglycerol and cholesterol and better insulin sensitivity than control littermates [41,42]. Furthermore, apolipoprotein E (*Apoe*)-deficient mice which also lack FABP4 are less prone to develop atherosclerosis than *Apoe*-deficient mice with FABP4 [43]. Several studies report that FABP4 is the main responsible factor for fatty acids phagocytosis by macrophages, leading to the formation of foam cells [44]. TLR4 agonists such as lipopolysaccharide trigger the transactivation of the c-Jun N-terminal kinase (JNK) and FABP4, which leads to the recruitment of c-Jun on the *FABP4* promoter and launches a positive feedback loop to accumulate FABP4 [45]. Concomitantly, FABP4 and TLR4 activation promote the nuclear factor Kappa-B (NF-κB) signaling pathway [46]. More than a fatty acid lipocalin implicated in lipids trafficking, recent advances show FABP4 as a pro-inflammatory adipokine.

Like RBP4, FABP4 has all the traits of a harmful adipokine promoting metabolic diseases. While further studies are still needed to complete FABP4 role in type 1 diabetes and in inflammation, this deleterious adipose-derived lipid-binding protein may be a new target to treat type 2 diabetes.

## 4. APOD

APOD was first identified by McConathy and Alaupovic in 1973 on HDL surface. It is a 169 amino acids protein with two glycosylation sites responsible for its varying molecular weight ranging from 20 to 32 kDa [47]. APOD was reported as produced by the adipose tissue in mice and humans [48,49] but also by a large variety of tissues [50]. It is able to associate with lecithin-cholesterol acyl-transferase (LCAT) and Apolipoproteins A-I or A-II, and transport cholesterol onto HDL particles [51,52]. As a lipocalin, APOD binds arachidonic acid and, with a lower affinity, progesterone and pregnenolone [53,54]. *APOD* gene polymorphisms were found in several populations, some of these variants being associated with a modified risk of Alzheimer’s disease or type 2 diabetes [55,56].

In a cohort of moderate to severely obese women, Desmarais et al. showed that, despite a differential expression of *APOD* between omental, mesenteric and round ligament adipose depots, APOD protein levels in fat tissues are positively correlated with the insulin sensitivity index QUICKI and negatively correlated with circulating levels of IL-6 and TNFα. In abdominal adipose tissue, higher APOD protein levels are associated with lower BMI, waist circumference and less insulin resistance [48]. However, *APOD* expression in cells extracted from amniotic fluid of obese women during pregnancy is nine-fold higher than in lean pregnant women [57]. These variations of *APOD* mRNA level between obese non-pregnant women and obese pregnant women might be driven by estradiol regulation on *APOD* expression, as circulating estradiol concentration increases during pregnancy and downregulates *APOD* expression in breast cancer [58]. In gestational diabetes, APOD placenta and plasma levels are higher in women with gestational diabetes than women without, whether they are overweight, obese [59] or not [60].

*Apod*-deficient mice display an altered lipidomic profile with elevated plasma triglycerides and insulin, without changes in glucose homeostasis evaluated by glucose tolerance test [61]; whereas mice overexpressing human *APOD* develop insulin resistance and hepatic steatosis with aging, but not with obesity or inflammation [62,63]. Hepatic steatosis after human *APOD* overexpression in mice is due to an upregulation of PPARγ, leading to an increased formation of lipid droplets in hepatocytes [64]. Obese and diabetic models such as the *ob*/*ob* and *db*/*db* mice exhibit lower levels of plasma APODthan heterozygous control littermates [65,66], which might be due to an altered interaction between APOD and the leptin receptor [65,67]. Similar results were obtained in obese male rats, in which *Apod* expression was downregulated in the reproductive system after a high fat diet [68].

On another hand, *Apod*-deficient mice have a higher sensitivity to oxidative stress in brain and develop locomotor and learning issues, while an overexpression of human *APOD* in brain rescued this phenotype [69]. In murine NIH/3T3 cells, a study tested various stress conditions in order to evaluate *Apod* regulation. The authors show that hydrogen peroxide H_2_O_2_ upregulates *Apod* expression in these adipocytes [70]. Taken together, these data suggest antioxidant properties of APOD.

Such characteristics have made APOD a debated topic in the field of obesity, diabetes and cardiovascular diseases. Further studies are needed to clarify the role of APOD in metabolic diseases, especially in obesity and type 2 diabetes, including men. So far, only the neuroprotective and antioxidant properties of APOD are known [11,71].

## 5. LCN2

The human neutrophil gelatinase-associated lipocalin NGAL, siderocalin, murine 24p3 protein or LCN2 was first identified in the early 1990s as a small protein secreted by neutrophils and bound to the matrix metalloproteinase (MMP)-9 [12,72]. It is a 25 kDa lipocalin able to bind iron and siderophores, bacterial iron-binding metabolites which captures iron from its host transporter proteins, therefore inducing a bacteriostatic effect [73]. This unique characteristic among the lipocalin family explains the wide expression pattern of LCN2 in pathogen-exposed tissues such as salivary glands, stomach, appendix, colon, trachea, lung, heart, kidney, uterus, prostate, blood cells and bone marrow [74,75,76].

Recent studies have demonstrated that plasma LCN2 is elevated in patients with acute myocardial infarction and is associated with poor outcomes like cardiogenic shock or increased risk of mortality [77,78]. As LCN2 levels also positively correlate with kidney disease, it has been proposed as a novel biomarker for cardiac event and as a predictive marker for acute kidney injury following a cardiac surgery [79].

*LCN2* is highly expressed in liver [80] and adipose tissue [81], so its role in metabolic diseases has been explored. In human cohort studies, plasma level and mRNA expression of LCN2 in liver and adipose tissue are higher in obese and diabetic subjects compared with lean individuals [80,82,83]. Despite higher levels in men than women, this difference remains after adjustment for sex [80]. Wang et al. also reported a positive correlation between serum LCN2 and indexes of adiposity (e.g., BMI, waist circumference, body fat percentage) and indexes of insulin resistance (e.g., HOMA-IR, fasting insulin and glucose levels) but this is not confirmed by another study [82]. LCN2 plasma levels and expression in adipose tissue are also positively associated with inflammatory markers in humans, hs-CRP and IL-6 [80,82]. An 8-week rosiglitazone treatment decreases the plasma levels of CRP as well as LCN2. Petropoulou et al. showed that circulating LCN2 increases after a meal in lean individuals, but not in obese subjects, and this increase correlates with a reduced hunger feeling [84], suggesting an anorexigenic effect of LCN2 in lean humans. This was supported by human LCN2 administration in monkeys, which displayed an acute 21% reduced food intake without short-time side effects.

However, these observations are in contradiction with a recent study, which shows an increase of food intake in *Lcn2*-overexpressing mice [85]. The authors also demonstrate high insulin levels in plasma from these mice. Moreover, *db*/*db* mice and streptozotocin-treated mice have higher circulating LCN2 and mRNA levels in liver and adipose tissue than wild-type littermates [80,86].

Several publications are in contradiction regarding the role of LCN2 in insulin resistance in mice. The same year, while Law et al. reported an improved insulin sensitivity in *Lcn2*-deficient mice compared to wildtype littermates [87], Guo et al. showed that *Lcn2* deficiency in mice promotes insulin resistance [88]. This paradox was studied by another group which reported increased insulin levels and increased insulin sensitivity in *Lcn2*-overexpressing mice [89]. The authors gave evidence of an improved β-cell function and insulin secretion in these mice, thereby increasing plasma insulin, a hallmark of insulin resistance. Moreover, LCN2 promotes muscle differentiation, improves thermogenic function of brown adipose tissue (BAT) and triggers beiging in white adipose tissue [90,91]. Interestingly, activation of BAT consequently to LCN2 signal is independent of the classical β-adrenergic/AMPK pathway and involves p38/MAPK [92].

Further studies are needed to get an overview of all LCN2 effects on metabolic pathways. While LCN2 is considered as a predictive marker for cardiac outcomes, its importance in immunity response has also been considered. Regarding its role in the onset of diabetes, a novel hypothesis is emerging: *LCN2* may be upregulated during the earliest stage of diabetes to promote β-cell function and insulin sensitivity but this protective mechanism is eventually overwhelmed when diabetes settles.

## 6. LCN14

LCN14 was first identified in mice as an odorant binding protein (OBP) with two isoforms, 2a and 2b. In mice, it shares a 50% homology with the OBP-1 variant produced by hepatocytes, also known as lipocalin-13. LCN14 is mainly expressed in white adipose tissue and poorly expressed in brown adipose tissue. In high fat diet-induced obese mice as well as diabetic *db*/*db* mice, LCN14 expression in the adipose tissue is lower than in wildtype, normal chow diet-fed mice [93].

Lee et al. showed that LCN14 is a beneficial adipokine which is able to enhance insulin sensitivity of liver and adipose tissue [93]. Overexpression reduces isoprenaline induced lipolysis and therefore might limit hepatic gluconeogenesis [93].

Murine LCN14 shares a 60% homology with human OBP2a and OBP2b [94,95], which also share similarities with rat OBP2. These two human isoforms, similar at 90%, display ability to bind olfactory molecules in the nasal tractus but so far, their expression in the adipose tissue has not been studied [95,96].

Overall, the putative positive role of LCN14 in metabolic homeostasis now deserves additional studies.

## 7. APOM

APOM was discovered in 1999 by Xu and Dahlbäck as a novel apolipoprotein mainly secreted by liver and kidney, and at lower level by adipose tissues [13]. Plasma APOM is mostly found as monomers on HDL, yet a recent study gave evidence of an in vitro homo- or heterodimerization of APOM when HDL particles are incubated in high-glucose buffer, though the authors could not find any APOM dimers in serums of diabetic subjects [97]. The human APOM protein is a lipocalin with one glycation site, so that its molecular weight varies from 21 to 26 kDa, but the murine APOM protein has an uncommon β-barrel lipocalin structure formed by only seven β-strands and no glycation site [98]. Whether this rare structure influences the murine APOM binding and affinity for its ligands is still an open question. In plasma, APOM is the main chaperone for S1P on HDL, also it binds retinol, retinoic acid, oxidized phospholipids and some fatty acids [13]. S1P is a bioactive lipid with diverse effects due to its five different G-protein coupled receptors, S1P1 to S1P5. S1P1 to S1P3 are widely distributed, while S1P4 is almost exclusively expressed in the lymphatic system and S1P5 in the nervous system. When bound to S1P1 or S1P3, S1P displays protective effects against insulin resistance and atherosclerosis, whereas S1P bound on S1P2 promotes insulin resistance and atherosclerosis [99,100]. Moreover, some genetic variants of the *APOM* gene reported in the Chinese population associate low plasma APOM levels with a higher risk of developing type 2 diabetes [101]. Therefore, the APOM/S1P complex as well as free APOM might be interesting targets in the metabolic diseases research area.

Plasma APOM levels are lower in obese, metabolic syndrome or type 2 diabetic subjects, and in women with gestational diabetes compared to lean, non-diabetic individuals. *APOM* expression in adipose tissue and liver is concordant with circulating APOM levels [60,102,103]. However, in type 1 diabetic patients, plasma APOM levels are not different to plasma levels from non-diabetic controls, in two cohort studies [104,105]. When submitted to a hypocaloric diet inducing a significant weight loss and improvements in the metabolic profile, APOM expression and secretion from the adipose tissue of obese individuals increase during calorie restriction [102]. Further details on human cohorts are reviewed in [14].

Animal studies provide further insight into the role of APOM in metabolic diseases. Goto-Kakizaki rats, a non-obese, spontaneous model of diabetic rats display better insulin sensitivity when overexpressing *Apom*, concomitantly with an increased insulin secretion and glucose utilization [106]. Concordant results were obtained in *Apom*-deficient mice which present impaired insulin sensitivity, high plasma inflammatory markers such as IL-6 or IL-1β, and hepatic NF-κB [99,107]. Yao et al. also recently demonstrated that APOM is expressed in murine and human macrophages, interacts with the scavenger receptor BI to promote cholesterol efflux from the phagocytic cells and protects against atherosclerosis [108,109]. More, *Apom*-deficient mice display blood hypertension and cardiac hypertrophy compared to wildtype animals [110].

Taken together, these observations indicate that APOM might be beneficial in the field of cardiac and metabolic diseases. However, APOM binding to S1P may complicate the elucidation of APOM part on insulin sensitivity. Christoffersen et al. showed that *Apom*-deficient mice have a more active BAT and improved insulin sensitivity [111], a phenotype also observed in *S1P2*-deficient mice [112]. Whether APOM only acts as a chaperone for S1P or is by itself a beneficial cardiometabolic adipokine is still a matter of debate, the latter being more and more considered.

## 8. Clinical Perspectives

Targeting RBP4, FABP4 or their signaling pathway to treat diabetes or atherosclerosis has been a profuse topic for the last few years. Torabi et al. designed calcium-phosphate nanoparticles conjugated with a single-stranded DNA aptamer which specifically binds to RBP4. This engineered nanoparticle ligand inhibits RBP4 binding to its natural ligand, vitamin A [113]. Other groups have tested already commercially available drugs, used for their antidiabetic properties, and studied the variations of RBP4 and FABP4 levels: both sodium-glucose cotransporter 2 (SGLT2) and dipeptidyl peptidase 4 (DDP4) inhibitors in combination with metformin can decrease RBP4 and FABP4 levels [114], while metformin alone reduces RBP4 and FABP4 levels and inflammatory markers [115]. Angiotensin II receptor blockers (namely candesartan, olmesartan, valsartan and telmisartan) [116], atorvastatin [117] and omega-3 fatty acids (namely eicosapentaenoic and docosahexaenoic acids) [118] induce a 8–20% decrease of FABP4 plasma levels in diabetic, hypertensive or dyslipidemic patients, whereas treatment of human adipocytes in vitro with bisphenol A induces an increase of *FABP4* expression [119].

So far, two groups have successfully used monoclonal antibodies directed against FABP4 which improve insulin sensitivity or attenuate inflammation [120,121], while others have developed novel small inhibiting molecules which stick in FABP4 hydrophobic pocket as molecular lures and prevent its effects [122,123].

## 9. Conclusions and Perspectives

Adipose tissue secretes a large range of bioactive molecules, including lipids. In healthy states, these factors maintain metabolic homeostasis. When adipose tissue expands, many exhibit increased secretion and are implicated in metabolic disturbances, and some that show lower secretion might contribute to maintain “healthy” metabolism. Among the six lipid-binding adipokines discussed in this review, most display effects closely associated with their ligand’s biological properties. In addition, these lipid-binding proteins have a large diversity of ligands, from fatty acids to sphingolipids, odorant volatile molecules to haeme metabolites. To understand the complex mechanisms by which those lipid-binding adipokines exert their effect, identification of other ligands warrants further investigations. Consideration should also be given to the possibility that these adipose-derived lipid-binding proteins have ligand-independent roles in metabolic diseases.

## Figures and Tables

**Figure 1 ijms-22-10460-f001:**
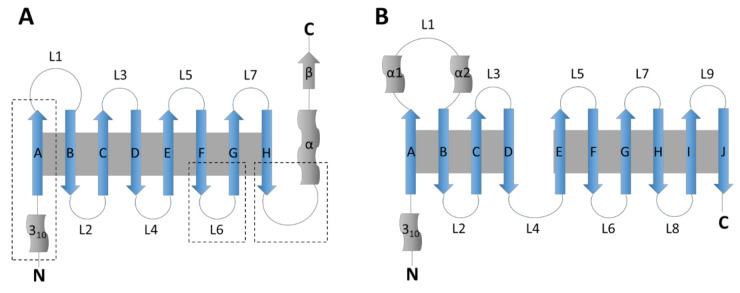
Schematic structures of the lipocalin and fatty acid binding protein family (adapted from [4]). (**A**) Lipocalin structure. The β-strands forming the β-barrel are represented as arrows labeled A–H and the loops as circular lines labeled L1–L7. The short N-terminus 3_10_-helix and the C-terminus α-helix and short β-strand are also represented. Grey background represents hydrogen bonds between strands. Dotted squares surround the three structurally conserved regions of kernel lipocalins. (**B**) Fatty acid binding protein structure. The β-strands are represented as arrows labeled A–J and the loops as circular lines labeled L1–L9. Two conserved α-helixes in loop L1 are shown as α1 and α2. The N-terminus 3_10_-helix and the C-terminus are also represented. Grey background represents hydrogen bonds between strands.

**Table 1 ijms-22-10460-t001:** Adipose-derived lipid-binding proteins features.

Gene Symbol/Aliases	Gene Location	Tissue Expression	Ligands	Biological Functions	Review
RBP4, Retinol binding protein	HSA 10q23.33 MMU 19	Adipose tissue, liver	Retinol retinoic acid	Vitamin A transport Inflammation Glucose homeostasis	[8,9]
FABP4/aP2, A-FABP, ALBP	HSA 8q21.13 MMU 3	Adipose tissue, immune cells (macrophages)	Fatty acids	Lipogenesis/Lipolysis Inflammation	[10]
*APOD/GCDFP-24*	*HSA 3q29* *MMU 16*	*Brain, mammary gland, lacrimal gland, pancreas, kidney, heart, adipose tissue, liver, intestine*	*Arachidonic acid* *Sphingomyelin* *Progesterone* *Pregnenolone* *Retinoic acid* *Bilirubin*	*Immunity/Inflammation* *Neuronal function* *Embryonic development* *Sexual development* *Antioxidant* *Cholesterol transport* *Glucose homeostasis*	[11]
*LCN2/NGAL,* *siderocalin, murine 24p3 protein, rat α2-microglobulin related protein*	*HSA 9q34.11 MMU 2*	*Kidney, bone marrow, liver, adipose tissue, immune cells, salivary gland, stomach, intestine, trachea, lung, heart, uterus, prostate*	*Iron, siderophores* *Leukotrienes* *Retinoic acid* *Unsaturated fatty acids*	*Immunity/Inflammation* *Renal and cardiac function* *Glucose homeostasis*	[12]
**OBP2a/2b/LCN14**	**HSA 9q34.2** **MMU 2**	**Nasopharynx, lachrymal gland, prostate, mammary gland, adipose tissue (mouse)**	**Odorant molecules** **(e.g., aldehydes, carboxylic acids)**	**Odors detection** **Pheromones signaling** **Glucose homeostasis**	-
**APOM/G3a, NG20**	**HSA 6p21.33** **MMU 17**	**Liver, kidney, adipose tissue, brain (blood–brain barrier)**	**S1P** **Saturated fatty acids** **Retinol, retinoic acid** **Oxidized phospholipids**	**Glucose homeostasis** **Inflammation**	[13,14]

RBP4: Retinol-binding protein 4; FABP4: Fatty acid-binding protein 4; aP2: adipocyte protein 2; A-FABP: adipocyte fatty acid binding protein; ALBP: adipocyte lipid-binding protein; APOD: Apolipoprotein D; GCDFP-24: Gross cystic disease fluid protein 24; LCN2: Lipocalin-2; NGAL: Neutrophil gelatinase-associated lipocalin; LCN14: Lipocalin-14; OBP: Odorant binding protein; APOM: Apolipoprotein M; Chr.: chromosome; S1P: sphingosine-1-phosphate. Regular type indicates adipokines associated to unhealthy phenotype, italics indicate adipokines described as detrimental or beneficial depending on the report, and bold type indicates adipokines reported as associated to healthy phenotype.

## Data Availability

Not applicable.

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
