# Peer review of "Adipose-Derived Lipid-Binding Proteins: The Good, the Bad and the Metabolic Diseases"

_ijms, 2021, doi:10.3390/ijms221910460_

Round 1
Reviewer 1 Report
The paper by Frances et al. is a narrative review discussing the physiology and pathophysiology of six adipose-derived lipid binding proteins.
The paper is clearly written and easily understandable. The Authors report experimental and clinical evidence regarding each of the six adipose-derived proteins with good detail and highlighting the favourable and deleterious effects of their action.
The paper is interesting and up-to-date. The reference list is complete.
Specific comment
1.
Abstract
I would mitigate the dichotomy between good and bad adipokines in the text.
The sentence “Most of the adipokines are considered as deleterious, due to their pro-inflammatory, pro-atherosclerotic or pro-diabetic properties, while only a few would be designated as beneficial adipokines, “ should rephrased, as all of them appear to be useful. What is deleterious is their increase (i.e. leptin) or decrease (i.e. adiponectin) under pathological conditions.
2.
In line 39, after …diseases and diabetes, I would also add …and cancer.
Author Response
The paper by Frances et al. is a narrative review discussing the physiology and pathophysiology of six adipose-derived lipid binding proteins.
The paper is clearly written and easily understandable. The Authors report experimental and clinical evidence regarding each of the six adipose-derived proteins with good detail and highlighting the favourable and deleterious effects of their action.
The paper is interesting and up-to-date. The reference list is complete.
Reply: Thank you for the nice summary - no further comments to add.
Specific comment
Query 1.Abstract
I would mitigate the dichotomy between good and bad adipokines in the text.
The sentence “Most of the adipokines are considered as deleterious, due to their pro-inflammatory, pro-atherosclerotic or pro-diabetic properties, while only a few would be designated as beneficial adipokines, “ should rephrased, as all of them appear to be useful. What is deleterious is their increase (i.e. leptin) or decrease (i.e. adiponectin) under pathological conditions.
Reply: We have revised the Abstract to tone down the presentation of the adipokines.
Query 2.
In line 39, after …diseases and diabetes, I would also add …and cancer.
Reply: The text has been modified accordingly.
Reviewer 2 Report
In the presented manuscript, Frances et al. provide a review of lipid-binding adipokines. Overall, the manuscript is well written and logically organized to cover the topic of great scientific as well as clinical importance. That said, there are several issues that need the author's attention:
- Table 1. It would be advantageous if official gene/protein names are used and highlighted (e.g., in bold) and the synonyms are put into brackets. For instance, the LCN14 is not an official gene/protein symbol and should be put into aliases only. As for the genomic localization, please consider using the standardized chromosome nomenclature, e.g., FABP4 / HSA 8q21.13 / MMU 3. The lacrimal gland is misspelled.
- Adding information about whether genetic variation(s) in the genes coding for the individual adipokines have been found to affect related (or unrelated) phenotypes could complete the presented picture.
- While the parts describing the first three discussed adipokines follow a similar structure with a brief and informative concluding summary, this is lost in the parts describing FABP4, ApoD, LCN2, OBP2a, and ApoM.
- In the section focused on LCN2, the contradiction relating to the effect of Lcn2 deficiency on insulin resistance is introduced, but the following sentences (lines 276-281) do not clarify the situation and do not reflect the literature following the original findings.
- The English spelling, usage and style needs a check and, at places, slight improvement, e.g., line 38 "insulin-resistance", line 40 "some of which having", line 62 "two alpha-helix", line 200 "was reported produced" etc.
Author Response
In the presented manuscript, Frances et al. provide a review of lipid-binding adipokines. Overall, the manuscript is well written and logically organized to cover the topic of great scientific as well as clinical importance.
Reply: Thank you for the suggestions below to improve the review.
That said, there are several issues that need the author's attention:
Query1: Table 1. It would be advantageous if official gene/protein names are used and highlighted (e.g., in bold) and the synonyms are put into brackets. For instance, the LCN14 is not an official gene/protein symbol and should be put into aliases only. As for the genomic localization, please consider using the standardized chromosome nomenclature, e.g., FABP4 / HSA 8q21.13 / MMU 3. The lacrimal gland is misspelled.
Reply: Thank you to point this out. The table has been modified accordingly, as well as the mispelling.
Query2: Adding information about whether genetic variation(s) in the genes coding for the individual adipokines have been found to affect related (or unrelated) phenotypes could complete the presented picture.
Reply: We have expanded the text regarding genetics
Query3: While the parts describing the first three discussed adipokines follow a similar structure with a brief and informative concluding summary, this is lost in the parts describing FABP4, ApoD, LCN2, OBP2a, and ApoM.
Reply: We added a concluding summary to the sections describing apoD, LCN2, LCN14 and apoM.
Query4: In the section focused on LCN2, the contradiction relating to the effect of Lcn2 deficiency on insulin resistance is introduced, but the following sentences (lines 276-281) do not clarify the situation and do not reflect the literature following the original findings.
Reply: We have revised the text to summarize most of the reports regarding the link between LCN2 and insulin resistance. We hope to have fulfilled the reviewer’s query.
Query5: The English spelling, usage and style needs a check and, at places, slight improvement, e.g., line 38 "insulin-resistance", line 40 "some of which having", line 62 "two alpha-helix", line 200 "was reported produced" etc.
Reply: Thank you to point this out. The text has been modified accordingly.
Reviewer 3 Report
This review is focused on adipose-derived lipid-binding proteins and their association with metabolic diseases. The manuscript is written on the basis of 120 references (about 50 from the last five years). In my opinion this manuscript is interesting but suffers from some flaws.
- The lack of main conclusion of this review (please add the conclusion section).
- It will be good summarize the good and the bed functions of these six adipokines in table of figure. This will make the text easier to understand.
- Line 100 (iii)?
- Fatty acid-binding protein 4 (FABP4) is also known as adipocyte FABP (A-FABP).
- I propose to include the clinical perspectives in the pharmacological or diagnostic fields,
Author Response
This review is focused on adipose-derived lipid-binding proteins and their association with metabolic diseases. The manuscript is written on the basis of 120 references (about 50 from the last five years). In my opinion this manuscript is interesting but suffers from some flaws.
Reply: Thank you for the valuable comments and suggestions. Please see detailed answer below:
Query1: The lack of main conclusion of this review (please add the conclusion section).
Reply: We now provide a conclusion (and perspectives) section that summarizes the content of the review.
Query2: It will be good summarize the good and the bed functions of these six adipokines in table of figure. This will make the text easier to understand.
Reply: The adipo-lipocalines are presented in the same order as in the main text. Actually we started with adipokines clearly reported as deleterious and finished with the adipokines associated with healthy metabolism. In between are the adipokines either good or bad depending on the reports. We have revised the Table by including a font code regarding the type of adipokines, from bad (regular) to beneficial (bold) ones. The legend of the table is implemented accordingly.
Query3: Line 100 (iii)?
Reply: For a sake of clarification the phrasing was changed.
Query4: Fatty acid-binding protein 4 (FABP4) is also known as adipocyte FABP (A-FABP).
Reply: Thank you to point this oversight out. The text has been modified accordingly.
Query5: I propose to include the clinical perspectives in the pharmacological or diagnostic fields,
Reply: We have revised the text to provide a clinical perspectives section (section 8).